Vertical escape tactics and movement potential of orthoconic cephalopods

Peterman David J. David.Peterman@utah.edu
Ritterbush Kathleen A.
Department of Geology and Geophysics, University of Utah , Salt Lake City, UT , United States
De Baets Kenneth
Electronic publication date: 2021 Jul 16
Publication date: 2021
Volume: 9
Electronic Location ID: e11797
Received 2021 Apr 21; Accepted 2021 Jun 25
Copyright: © 2021 Peterman and Ritterbush
Copyright year: 2021
Copyright holder: Peterman and Ritterbush
License: This is an open access article distributed under the terms of the Creative Commons Attribution License, which permits unrestricted use, distribution, reproduction and adaptation in any medium and for any purpose provided that it is properly attributed. For attribution, the original author(s), title, publication source (PeerJ) and either DOI or URL of the article must be cited.
License URL: https://creativecommons.org/licenses/by/4.0/

Keywords: Cephalopod, Ammonoidea, Nautiloidea, Functional morphology, Biomechanics, 3D Printing, Hydrostatics, Hydrodynamics, Orthocone, Paleobiology

Funding: National Science Foundation Award #1952756 This work was supported by the National Science Foundation (Award #1952756). The funders had no role in study design, data collection and analysis, decision to publish, or preparation of the manuscript.

==============================
Measuring locomotion tactics available to ancient sea animals can link functional morphology with evolution and ecology over geologic timescales. Externally-shelled cephalopods are particularly important for their central roles in marine trophic exchanges, but most fossil taxa lack sufficient modern analogues for comparison. In particular, phylogenetically diverse cephalopods produced orthoconic conchs (straight shells) repeatedly through time. Persistent re-evolution of this morphotype suggests that it possesses adaptive value. Practical lateral propulsion is ruled out as an adaptive driver among orthoconic cephalopods due to the stable, vertical orientations of taxa lacking sufficient counterweights. However, this constraint grants the possibility of rapid (or at least efficient) vertical propulsion. We experiment with this form of movement using 3D-printed models of Baculites compressus, weighted to mimic hydrostatic properties inferred by virtual models. Furthermore, model buoyancy was manipulated to impart simulated thrust within four independent scenarios (Nautilus-like cruising thrust; a similar thrust scaled by the mantle cavity of Sepia; sustained peak Nautilus-like thrust; and passive, slightly negative buoyancy). Each model was monitored underwater with two submerged cameras as they rose/fell over ~2 m, and their kinematics were computed with 3D motion tracking. Our results demonstrate that orthocones require very low input thrust for high output in movement and velocity. With Nautilus-like peak thrust, the model reaches velocities of 1.2 m/s (2.1 body lengths per second) within one second starting from a static initial condition. While cephalopods with orthoconic conchs likely assumed a variety of life habits, these experiments illuminate some first-order constraints. Low hydrodynamic drag inferred by vertical displacement suggests that vertical migration would incur very low metabolic cost. While these cephalopods likely assumed low energy lifestyles day-to-day, they may have had a fighting chance to escape from larger, faster predators by performing quick, upward dodges. The current experiments suggest that orthocones sacrifice horizontal mobility and maneuverability in exchange for highly streamlined, vertically-stable, upwardly-motile conchs.

Introduction

A phylogenetically-diverse array of cephalopod mollusks produced straight conchs (orthocones) throughout geologic time, but their ecological contributions to marine systems are unclear. The evolutionary contexts of these animals are well documented: iconic spiral conchs of nautilids (extant) and ammonoids (extinct) are heavily-derived forms that follow early success by orthocone relatives (Holland, 1987). Orthoconic nautiloids were globally distributed and diverse in the Paleozoic, yielding hundreds of fossil genera (Teichert et al., 1964). A branch of orthocerid nautiloids gave rise to orthoconic bactritoids, and eventually to ammonoids (Erben, 1966; Monnet, Klug & De Baets, 2015). Evolution of tightly-coiled ammonoid conchs in Early Devonian seas (by increased exogastric curvature; Klug & Korn, 2004; Kröger, 2005; Kröger & Mapes, 2007; Monnet, De Baets & Klug, 2011) may have radically increased horizontal mobility via hydrostatic and hydrodynamic features (aligning jet thrust to the animal’s center of mass while simultaneously improving lateral streamlining; Klug & Korn, 2004). While coiled nautiloids evolved several times throughout the Paleozoic (e.g., mostly within Tarphycerida, Lituitida, Nautilida), the diversity and abundance of coiled ectocochleates relative to orthocones increased during this Devonian nekton revolution (Klug et al., 2010). Yet orthoconic nautiloid lineages commonly persisted until the Late Triassic and rarely into the Early Cretaceous (Doguzhaeva, 1994). Furthermore, orthoconic ammonoids repeatedly originated from planispiral ancestors through the Mesozoic (e.g., heteromorph species within Triassic Choristoceratidae, Jurassic Spiroceratidae, Late Jurassic/Early Cretaceous Bochianitidae, and Cretaceous Baculitidae; Wiedmann, 1969; Wright, Callomon & Howarth, 1996; Hoffmann et al., 2021). The persistence and intermittent appearance of orthoconic forms suggests that this morphotype retained adaptive value, despite proliferation of planispiral conchs among these animals’ contemporaries. However, the adaptive value, functional morphology, and ecology of orthoconic cephalopods are poorly understood. We aim to investigate these properties through hydrostatic and hydrodynamic analyses using the Cretaceous baculitid, Baculites compressus (North America, late Campanian), as a test case.

Hydrostatics of orthoconic cephalopods

Hydrostatic analyses suggest that orthoconic cephalopods assumed vertical orientations at rest (Trueman, 1941; Westermann, 1977, 1996; Peterman et al., 2019; Peterman, Barton & Yacobucci, 2019). Mass would be anteriorly distributed near the body chamber (housing the animal’s soft body) due to the overlying air-filled chambers (the phragmocone buoyancy apparatus). A static orientation occurs when the total center of mass is vertically aligned under the center of buoyancy (Hoffmann et al., 2015). When these centers are forced out of alignment, a restoring moment proportionate to their separation will act to return them to their static, equilibrium condition (Peterman et al., 2019; Peterman et al., 2020a; Peterman et al., 2020b). Mineral deposits (i.e., cameral and endosiphuncular deposits) common among some Paleozoic nautiloid species would have influenced hydrostatics in some way. These deposits are formed syn vivo (Seuss et al., 2011; Pohle & Klug, 2018; for contrasting views see Mutvei, 2018), very disparate in structure (Teichert, 1933; Flower, 1955a; Fischer & Teichert, 1969), and help characterize several higher taxa of nautiloids (King & Evans, 2019). Though diverse clades had such mineral deposits in their shells, they have been generally regarded as counterweights that facilitated more horizontal postures in the water column (Schmidt, 1930; Flower, 1955b; Westermann, 1977; Holland, 1987; Crick, 1988; Chamberlain, 1993; Barskov et al., 2008). However, recent studies suggest these counterweights may have altered restoring moments and dynamic conch orientation, while the static orientation would likely remain near vertical (Peterman, Barton & Yacobucci, 2019). More detailed simulations are required to understand the degree to which these structures reduce hydrostatic stability.

Hydrostatic and hydrodynamic relationships (Peterman et al., 2019; Peterman, Barton & Yacobucci, 2019) do not readily support interpretations of orthoconic cephalopods as swift horizontal swimmers, akin to extant squid (Tsujita & Westermann, 1998). First, the source of jet thrust (i.e., the hyponome) is aligned vertically with the centers of buoyancy and mass (Peterman et al., 2019), supporting that thrust energy would most efficiently be transmitted into upward vertical movement during jet propulsion. If thrust was applied in a horizontal direction, much energy would be lost to rocking since the source of jet thrust is situated much lower than these two hydrostatic centers (Peterman et al., 2019). Moreover, thrust perpendicular to the long axis of the conch would not be sufficient to orient the animal horizontally (for taxa lacking substantial counterweights; Peterman et al., 2019). These properties strongly constrained how orthoconic cephalopods would have interacted with their surroundings, fed, and evaded predators. The source of thrust relative to the mass distribution and vertically streamlined conch suggest that orthocones were adapted for improved vertical movement potential, but at the expense of horizontal mobility.

Orthoconic cephalopod paleoecology

Constraining likely locomotory functions for orthoconic cephalopods can lend context to a very diverse range of lineages that flourished throughout the Paleozoic and Mesozoic, and are known from nearly all marine paleoenvironments around the globe (Kennedy & Cobban, 1976; Wright, Callomon & Howarth, 1996; Kröger & Zhang, 2009; Kröger, Servais & Zhang, 2009). Orthoconic nautiloids are generally regarded as vertical migrants of the water column and/or demersal based on occurrence data, morphological characters, and taphonomic patterns (Kröger, Servais & Zhang, 2009). Species recovered from offshore sediments may have migrated vertically through pelagic landscapes of varied photic and oxygen zones; whereas species recovered from more coastal deposits may have remained in the neritic zone where demersal feeding would be available (Kröger, Servais & Zhang, 2009). This demersal lifestyle of feeding on benthic fauna is traditionally interpreted for orthocones (Frey, 1989; Brett & Walker, 2002; Barskov et al., 2008; Kröger & Zhang, 2009). The remarkable disparity of orthocone conchs (especially siphuncular characteristics; Fischer & Teichert, 1969; Barskov et al., 2008) suggests there were some fundamental and/or nuanced differences in their life habits. Therefore, either vertical migration or primarily demersal lifestyles are not ruled out in the appropriate settings.

Orthoconic ammonoids, in contrast, are generally found in neritic and epicontinental settings (Kennedy & Cobban, 1976; Wright, Callomon & Howarth, 1996). The depth range of baculitid ammonoids was around 50-100 m based on isotopic analyses of well-preserved shell material (Fatherree, Harries & Quinn, 1998; Lukeneder et al., 2010; Henderson & Price, 2012; Lukeneder, 2015; Sessa et al., 2015; Landman et al., 2018; Hoffmann et al., 2021). A demersal life habit is generally inferred from these analyses due to their isotopic similarity with the benthos (Landman et al., 2018; Ferguson et al., 2019; Hoffmann et al., 2021). Isotopic studies also suggest that some baculitids spent most of their lives at methane seeps, supporting a somewhat sedentary lifestyle (Landman et al., 2018; Rowe et al., 2020). However, baculitid associations with streamlined midwater swimmers, and occurrences in deposits lacking demersal taxa, suggest that these species could cope with life higher in the water column as well (Tsujita & Westermann, 1998; Landman, Cobban & Larson, 2012).

Key observations remain that suggest horizontal (or subvertical) modes were adopted by at least some species of orthoconic cephalopods. Life habit can be inferred from color patterns preserved on the conch. While some orthocones had color patterns around the entire circumference, others have patterns restricted to the dorsum, suggesting countershading in a nonvertical orientation (Packard, 1988; Westermann, 1998; Kröger, Servais & Zhang, 2009; Manda & Turek, 2015). These orientations can be explained by resting the soft body on the benthos (Flower, 1955c) or by the use of active locomotion. The former would require some amount of negative buoyancy and would only provide useful camouflage from above, while the latter would require sustained jet thrust of considerable magnitude in forms with lower hydrostatic stability (i.e., those with cameral or endosiphuncular deposits; Peterman, Barton & Yacobucci, 2019). Contrasting hydrostatic interpretations, aperture-forward, horizontal movement was inferred for the baculitid ammonoid, Sciponoceras, based on the adoral growth direction of a cirripede attached to the venter (Hauschke, Schöllmann & Keupp, 2011). Similar growth orientations are observed for epizoans on some nautiloid orthocones as well (Baird, Brett & Frey, 1989).

While a single mode of life should not be invoked for all orthoconic cephalopods, the practical challenges faced during the animals’ life can be constrained through empirical studies. The model ammonoid, Baculites compressus, can provide valuable insight for the vertical movement potential and kinematics of this hydrostatically-stable morphotype. The anatomy and propulsive capabilities likely varied within and between ammonoid and nautiloid groups; a wide range of experimental conditions allow broader interpretations for how this common conch morphotype may have functioned. Orthocone conch size, ornamentation, curvature, whorl section anatomy, body chamber proportion, and internal sculpture vary between clades; certain physical properties, too, could progressively differ for taxa with higher disparity from our investigated species. Therefore, we primarily limit our interpretations to orthoconic ammonoids and morphologically-similar nautiloids without substantial conch curvature or internal counterweights. The ease or difficulty in vertical movement, similar to the behavior observed in extant nautilids (Ward et al., 1984; Ward, 1987; O’Dor et al., 1993; Dunstan, Ward & Marshall, 2011) is of interest. Furthermore, the conditions required to vertically dodge larger, faster, predators with more speed-efficient modes of locomotion (undulatory vs jet propulsion; Webber & O’Dor, 1986; Anderson & Grosenbaugh, 2005; Neil & Askew, 2018) are explored with a range of predator analogues. An investigation of these capabilities can yield clues regarding the adaptive value of this enigmatic morphotype and its iterative recurrence in the fossil record.

Materials & methods

Three-dimensional motion tracking was performed on physical models of Baculites compressus in order to investigate various hydrodynamic properties of the orthocone morphotype. These models were constructed from the virtual hydrostatic model of Peterman et al. (2019) (Fig. 1A). The exterior portion of this model was isolated in the program Blender (Blender Online Community, 2017), then a soft body more closely resembling the baculitid reconstruction of Klug, Riegraf & Lehmann (2012) was fabricated and affixed to the aperture using the same software (Fig. 1B). The coiled embryonic shell (ammonitella) was ignored due to its very small scale (~0.7 mm; Landman, 1982). Instead, the model tapers to a point approximately 0.7 mm in diameter. The Baculites compressus model in the current study was designed to have positive or negative buoyant forces that simulate movement in the vertical directions using similar methods to Peterman et al. (2021a). Each model in the current study was constructed at a size of 57 cm (from the conch apex to distal ends of the arms). An external 3D model of Baculites compressus is stored in the morphosource.org database (ark:/87602/m4/359359).

Figure 1 Construction of a physical, 3D-printed model of Baculites compressus from a virtual hydrostatic model.

(A) Virtual model used to determine hydrostatic properties (modified from Peterman et al., 2019). (B) Virtual model with simplified internal geometry that allows for 3D printing. The total model mass was manipulated to impart an upward buoyant force, simulating downward thrust. The total center of mass (m) relative to the center of buoyancy (b) was maintained with an adorally placed counterweight and various internal voids. (C) Physical, 3D-printed model with tracking points placed at the distal ends of the arms and apex used for 3D motion tracking.

Vertical movement scenarios

Differences between the mass of water displaced and total model mass were computed to equal the forces produced during movement under four different scenarios: (1) Nautilus-like cruising thrust, (2) Nautilus-like cruising thrust scaled by the mantle cavity ratio of extant cuttlefish, (3) sustained maximum Nautilus-like thrust and (4) slightly negative buoyancy similar to extant Nautilus. For Scenario 1, the thrust of 0.015 N required for a 73 g Nautilus to overcome drag at its maximum velocity (Chamberlain, 1987) was scaled by the mass of the water displaced by the current orthocone model (212.209 g) to yield a target thrust value of 0.0436 N. The mass of water displaced is equal to organismal mass, assuming a neutrally buoyant condition. This Nautilus-like thrust was then scaled from the mantle cavity ratio of Nautilus (0.15; Wells & O’Dor, 1991) to the mantle cavity ratio of Sepia (0.25; Wells & O’Dor, 1991) yielding a target thrust of 0.0727 N for Scenario 2. The maximum propulsive thrust for Nautilus is a function of body size (Chamberlain, 1987). For Scenario 3, this maximum thrust was computed by substituting the mass of the Baculites compressus model (212.209 g) into the following formula reported by Chamberlain (1987):

(1) Tmax=0.0021(m)–0.103

Where Tmax is the maximum propulsive thrust in Newtons and m is the organismal mass in grams.

The model for Scenario 4 was made slightly negatively buoyant with a similar magnitude observed in extant Nautilus. Ward & Martin (1978) report residual masses (not relieved by buoyancy) for several wild-caught Nautilus. Focusing on larger individuals from their study (>300 g), the average residual mass was 1.76 g with an average total mass (including the soft body, conch, and chamber liquid) of 676.6 g (~0.26% of their organismal mass is not relieved by buoyancy). This corresponds to a residual mass of 0.552 g for the orthocone model. In order to manage error, the model mass for this scenario was lowered by 1 g. The model was then made neutrally buoyant by adding liquid into an anterior chamber with a syringe through a self-healing rubber cap. Once the model no longer sank or rose for ~30 seconds, it was considered neutrally buoyant. Then the residual mass of 0.5 g was added with water through the syringe to make the model negatively buoyant according to the computed value. This negatively buoyant scenario was chosen to assess the speed of descent for this vertically-streamlined morphotype, and basic swimming capabilities (e.g., vertical migration, pouncing).

Each target buoyant force depends on the mass of the water displaced by the models, which depends on density. Water density (1.000 g/cm3) was computed with a calibrated 100 ml pycnometer using water from where the motion tracking experiments were conducted (described below). Water conditions including temperature (~28 °C) and salinity were held constant (or nearly so) for measurements and all experiments.

Baculites compressus model construction

Each model is composed of three parts (Fig. 1B); air-filled voids, PLA (polylactic acid) plastic, and bismuth counterweights. Bismuth was chosen because of its high density, low melting point, and softness. Since the virtual models of these components are only digital volumes, their densities were determined with a calibrated 100 ml pycnometer to compute their masses. Virtual models of the counterweights were placed anteriorly (within the arm crown) with fixed positions and volumes for each model. After determining an appropriate counterweight volume (and therefore mass) for each model, the volumes of the internal voids were adjusted to yield the proper total mass for each model. The positions of the voids were iteratively altered to maintain the same total model mass, while imparting the same hydrostatic stability index (0.505) and apex-upward orientation inferred from the virtual hydrostatic model (Peterman et al., 2019). The virtual models were considered finished when their hydrostatic stabilities and thrusts matched their target values to the third and fourth decimal places, respectively.

Hydrostatic stability indices were computed from the total center of mass and the center of buoyancy. The center of buoyancy was computed from the model of the exterior interface (i.e., a model of the water displaced). This center and the centers of mass for each material of unique density (air, water, PLA plastic, bismuth counterweight), were computed in MeshLab (Cignoni & Ranzuglia, 2014). The total center of mass was computed with the following formula:

(2) M=∑(D∗mo)∑mo

Where M is the total center of mass in a principal direction, D is the center of mass of a single object in each principal direction, and mo is the mass of any object of unique density. All coordinates are measured relative to the same arbitrary datum placed at the center of the aperture.

The hydrostatic stability index (St) was computed from the following equation (after Okamoto, 1996):

(3) St=DBMV3

Where DBM is the distance between the center of buoyancy (B) and the center of mass (M), computed with the 3D theorem of Pythagoras. V is the organismal volume (equal to the volume of water displaced).

The mass distribution of the PLA plastic required to impart the desired hydrostatic stability index was computed with the following formula:

(4) DPLA=M(mPLA+mBi+mair+mwater)−(DBimBi)−(Dairmair)−(Dwatermwater)(mPLA)

Where DPLA is the location of the PLA center of mass from the datum in each principle direction. M is the total center of mass in a particular principle direction, m is the mass of a model component, and D is the center of mass of each model component in a particular principle direction. Subscripts denote each model component. Water is present only in the negatively buoyant model (1 g).

After each virtual model was completed, they were 3D printed in a vertical orientation with an Ultimaker S5 in four parts (in order to fit within the available print volume). Each part contained a watertight void (Fig. 1B) that was sealed during 3D printing. The model parts were all printed without support material due to their generally low overhang angles (<60°). Each of the four parts were chemically welded together with dichloromethane. High heat silicone molds of the counterweights were cast from 3D prints. The counterweights were cast form these molds by heating bismuth in a casting ladle with a propane torch, and evenly pouring the molten bismuth into the mold. The counterweights were further processed by using a metal file and silicon carbide paper until the desired volume and mass was reached. The counterweight masses slightly differed from their virtual counterparts (from 0.7% to 5% different) because they were used to compensate for subtle differences between the virtual and actual masses of the 3D printed parts. This modification minimized error of the target masses at the expense of error in hydrostatic stabilities. The hydrostatic stability indices of the physical models were recomputed with Eq. (3), and all model components (aside from air) were weighed in grams to the third decimal place in order to report error in stability and simulated thrust. Tracking points were painted on the apex and the distal end of the arms for each final model (Fig. 1C) to monitor their position with motion tracking.

Motion tracking experiments

An underwater camera rig was designed to record video footage of the orthocone models as they rose/fell in the water (Fig. 2A). Experiments were performed in a 2.1-m-deep section of the crimson lagoon—a 50 m lap pool at the University of Utah. The skeleton of the rig was constructed with two-inch PVC pipe with custom 3D-printed fittings for two camera mounts. Three steel weights were positioned at each end of the T-shaped rig and rubber mats were wrapped around each section to prevent the rig from slipping on the pool liner. Each camera mount consists of a GoPro Hero 8 Black camera in a waterproof case on top of a waterproof LED light (Fig. 2B). Each camera was oriented with the long axis in the vertical direction for improved field of view. Videos were recorded at 60 frames per second with a linear field of view and 4K resolution. Each model was held with extendable tongs until steady, then released for a total of 15 trials for the positively buoyant scenarios, and 8 trials for the negatively buoyant scenario. The relationship between model texture and velocity were assessed by coating the orthocone model with peak Nautilus-like thrust (Scenario 3) with hydrophobic silicone spray and performing an additional 15 trials. Sample video footage for Scenario 3 is stored in an online repository (https://doi.org/10.5281/zenodo.4776924).

Figure 2 Underwater camera rig used for 3D motion tracking.

(A) Schematic of the camera rig relative to the model (yellow) and release mechanism (green). The rig was made strongly negatively buoyant with three steel counterweights at the ends (purple). The wireframe shapes radiating from the cameras denote the approximate field of view. (B) Close-up view of waterproof camera and LED light with custom, 3D-printed attachments.

Dual video footage was imported into the 3D motion tracking software DLTdv8 (Hedrick, 2008) and semi-automatic tracking was used to mark the pixel locations of each tracking point (apex and arms). A calibration was performed in easyWand5 (Theriault et al., 2014) to transform the 2D pixel coordinates from each video into a single set of 3D coordinates in meters. The model itself was used for wand calibration, ensuring any 3D orientation of the model yielded its actual body length (57 cm). Three-dimensional data points higher in the water column were subject to minor image artifacts from light interacting with the water surface. These distortions, in addition to the increasingly oblique apparent angles of each model, yielded higher error in these regions. Calibrations with standard deviations of less than 2 cm were considered acceptable. The very fast frame rate (60 fps) caused fluctuations in velocity at lower time steps. This was remediated by using a moving average with a window of 11 time steps for the positively buoyant experiments. No moving average was used on the negatively buoyant experiment because only every 10th frame was used to compute velocity (due to very low velocities and long trial times).

Velocities were computed as a function of time for each model using the calibrated, 3D datapoints:

(5) Vi=(xi−xi−1)2+(yi−yi−1)2+(zi−zi−1)2(ti−ti−1)

Where V and t are velocity and time, and the subscripts i and i-1 refer to the current and previous time steps, respectively. The 3D theorem of Pythagoras was used to compute the total distance traveled in any x, y, z direction between time steps (which was mostly vertical). The video frame number was divided by the frame rate (59.94 frames per second) to compute time. Time zero for each trial was defined as the moment the release mechanism no longer contacted the model.

The curve fitting toolbox in MATLAB R2020a was used fit the velocity data for each model with an asymptotic equation in the form:

(6) Vfit=a−ae−bt

Where Vfit is the fit velocity and t is time. The term “a” is a coefficient denoting the velocity asymptote (i.e., the maximum velocity estimate given a particular thrust), and b is a coefficient that governs slope.

The relationships between hydrostatic stability and hydrodynamic movement were assessed by computing the maximum angle displaced from the vertical static orientation in any particular direction (θdv):

(7) θdv=cos−1((z2−z1)(x2−x1)2+(y2−y1)2+(z2−z1)2)

Where the subscripts 1 and 2 of the x, y, and z coordinates refer to the arm and apical tracking points, respectively.

Predator evasion potential

By knowing how velocity increases from stationary initial conditions (Eq. (6)), the time it takes to move one body length (tbl) or half a body length (tbl/2) can be computed:

(8) ΔP=Lbody=∫0tbl⁡a−ae−bt

When the change in position (∆P) is equal to the length of the body (Lbody; 57 cm), tbl can be computed by integrating Eq. (6) and determining the upper limit of integration. This limit was computed iteratively in MATLAB by increasing tbl until the integrated equation was equal to Lbody. For an orthocone to dodge a horizontally moving predator, jetting at the last moment would be ideal for a vertical escape. The minimum distance required (D) to dodge a predator attack at some incident velocity (Vp) was computed by multiplying tbl or tbl/2 by the predator’s velocity (Vp). Since the critical swimming speeds of extinct animals are difficult to determine, extant marine predators were used as analogues.

Results

Model hydrostatics and error

Each of the four orthocone models have the same centers of buoyancy because their external volumes are identical (Table 1). However, the centers of mass for each component of unique density differed in order to maintain a stability index of 0.505 as total mass varied between models (Tables 1 and 2). The mass of the bismuth counterweights slightly differed from their virtual counterparts to compensate for differences in mass between the virtual and physical PLA components. This mass difference reduced error in total mass by sacrificing accuracy in hydrostatic stability (Table 3). The differences in PLA mass were likely a result of differences in bulk density between each of the models. Even with identical slicer settings, the path of the extruder in each layer was slightly different between models, which contributed to error in PLA mass and density. Percent difference of hydrostatic stability and thrust between the virtual and physical models is reported on Table 3.

Table 1 Model centers of mass and buoyancy.

Local centers of mass for the model components (PLA plastic and bismuth counterweight), and total centers of mass and buoyancy for each model (1: Nautilus-like cruising thrust; 2: Nautilus-like cruising thrust scaled by the higher mantle cavity ratio of Sepia; 3: Nautilus-like peak thrust; 4: Slightly negatively buoyant). Only the x and z values are reported because the virtual model is perfectly symmetrical. All coordinates are measured relative to the same arbitrary datum (located in the center of the aperture).

	PLA	Bismuth	Total center of mass	Center of buoyancy	
Baculite
model	x (mm)	z (mm)	x (mm)	z (mm)	x (mm)	z (mm)	x (mm)	z (mm)	
1	−2.410	130.020	−0.032	−52.279	−1.810	84.070	−1.958	114.191	
2	−2.375	119.742	0.439	−57.934	−1.810	84.070	−1.958	114.191	
3	−2.052	100.828	0.245	−58.616	−1.810	84.070	−1.958	114.191	
4	−1.767	101.472	−2.366	−57.998	−1.810	84.070	−1.958	114.191	

Table 2 Masses (m) and volumes (V) for the virtual and physical model components.

	Virtual	Physical	
Baculite
model	VPLA
(cm3)	mPLA
(g)	VBi
(cm3)	mBi
(g)	mwd
(g)	mtotal
(g)	Mass def. (g)	mPLA
(g)	mBi
(g)	mglue
(g)	mwd
(g)	mtotal
(g)	Mass def. (g)	
1	124.319	155.275	5.651	52.490	212.209	207.765	4.444	157.875	49.737	0.101	212.209	207.713	4.496	
2	130.978	163.592	4.437	41.211	212.209	204.803	7.406	162.205	42.518	0.207	212.209	204.930	7.280	
3	126.935	158.542	2.018	18.745	212.209	177.287	34.922	158.529	18.612	0.355	212.209	177.496	34.713	
4	152.102	189.975	2.453	22.789	212.209	212.764	−0.555	187.063	24.875	0.206	212.209	212.144	-0.5*	
Note:

1: Nautilus-like cruising thrust; 2: Nautilus-like cruising thrust scaled by the higher mantle cavity ratio of Sepia; 3: Nautilus-like peak thrust; 4: Slightly negatively buoyant. PLA = 3D printed plastic; Bi = bismuth counterweight; wd = water displaced; Mass def. = mass deficiency required to impart the computed buoyant forces (Table 3); glue = the cyanoacrylate glue used to secure each counterweight. The residual mass in the negatively buoyant experiment (denoted with *) was not weighed, but rather its volume was inserted into the model with a syringe (~0.5 cm3).

Table 3 Virtual and actual hydrostatic stabilities (St) and thrusts (F), and computed percent errors.

Baculite
model	Virtual
St	Actual
St	St error (%)	Target
F (N)	Actual
F (N)	Thrust
error (%)	
1	0.505	0.492	−2.57	0.0436	0.0441	1.17	
2	0.505	0.499	−1.19	0.0727	0.0714	−1.77	
3	0.505	0.483	−4.36	0.3426	0.3405	−0.60	
4	0.505	0.454	−10.10	−0.0054	−0.0049	−9.42	
Note:

1: Nautilus-like cruising thrust; 2: Nautilus-like cruising thrust scaled by the higher mantle cavity ratio of Sepia; 3: Nautilus-like peak thrust; 4: Slightly negatively buoyant.

Motion tracking kinematics

The upward force on the physical models occurs on the center of buoyancy, not the location of the hyponome. However, this location of applied force is a reasonable assumption for these experiments because perfect upward-vertical thrust is still simulated. Furthermore, the actual source of thrust and the centers of buoyancy and mass are very close to being vertically aligned in living orthoconic cephalopods and in the current models. This alignment would allow thrust to be transmitted into primarily upward translation with little energy lost to rocking.

After releasing the models underwater, they primarily moved in the vertical directions with velocities proportionate to their simulated thrust values. Each of the trials were slightly skewed in the horizontal directions due to weak currents created by removing the release mechanism (Fig. 3). Additionally, the initial angles of some trials were slightly tilted from vertical, causing a few outlier trajectories (see Fig. 3A).

Figure 3 Three-dimensional positions from a common starting point for each model through time.

(A) Scenario 1: Nautilus-like cruising thrust. (B) Scenario 2: Nautilus-like cruising thrust scaled by the higher mantle cavity ratio of Sepia. (C) Scenario 3: Nautilus-like peak thrust. (D) Scenario 4: slightly negatively buoyant (~0.26% of mass not relieved by buoyancy).

During vertical movement, high hydrostatic stability prevented substantial displacements from vertical orientations (Fig. 4). The tracking points were about two degrees from true vertical in a static setting, and most trials remained under five degrees from vertical. Larger angles of displacement were usually the result of a tilted starting position (see Fig. 4A). The negatively buoyant model underwent larger displacements from vertical only at higher velocities (~−7 cm/s) and consistently tilted dorsum-upwards.

Figure 4 Maximum displacement angle in any direction from the vertical axis through time.

Each trial is distinguished by color. (A) Scenario 1: Nautilus-like cruising thrust. (B) Scenario 2: Nautilus-like cruising thrust scaled by the higher mantle cavity ratio of Sepia. (C) Scenario 3: Nautilus-like peak thrust. (D) Scenario 4: slightly negatively buoyant (~0.26% of mass not relieved by buoyancy).

The velocities between all trials of each positively buoyant model were remarkably similar (Figs. 5A–5C). However, the negatively buoyant model (Fig. 5D) was more sensitive to initial conditions (i.e., the subtle motion caused by holding the tongs before and during release) due to the relatively low force acting in the downwards direction. While Eq. (6) yields high R-squared values (>0.97), it underpredicts velocities at the start and end of the experiments, and slightly overpredicts velocities in the middle. This equation, however, provides a simple model to estimate orthocone swimming velocities using only a few terms (Table 4) and avoids overfitting the data. The velocity asymptote for each positively buoyant model is reported in Table 4. With sustained peak Nautilus-like thrust, the orthocone model reaches 1.2 m/s (2.1 body length per second) within one second from a stationary initial condition (Fig. 5C; Table 4). A sustained thrust during this narrow time window is a suitable assumption because it is on par with the propulsive period observed in extant Nautilus (~0.62–1.39 s; Chamberlain, 1987). While Eq. (6) reports asymptotic velocities, experiments with longer durations (over larger vertical distances) are required to determine the upper velocity limit (when hydrodynamic drag is equal to thrust). Coating the model with hydrophobic silicone yields little difference in velocity during the time window of the experiments (Fig. 5C). However, the velocity asymptote of Eq. (6) between the coating and uncoated models yields a difference in 23 cm/s, suggesting that this asymptote term is very sensitive. The negatively buoyant model is capable of reaching relatively high velocities (~15 cm/s) after about 20 s of uninterrupted sinking, falling from <2 m.

Figure 5 Velocity in the direction of movement as a function of time.

All trials are fit with an asymptote equation. (A) Scenario 1: Nautilus-like cruising thrust. (B) Scenario 2: Nautilus-like cruising thrust scaled by the higher mantle cavity ratio of Sepia. (C) Scenario 3: Nautilus-like peak thrust. A hydrophobic coating was applied to the original model to compare the influence of surface texture and friction drag. (D) Scenario 4: slightly negatively buoyant (~0.26% of mass not relieved by buoyancy).

Table 4 Velocities, travel times, and asymptote equation coefficients.

Uncertainty reflects bounds of 95% confidence intervals. The asymptotic velocity (in cm/s) is predicted by coefficient “a” of Equation 6. Coefficient “b” governs the slope. The maximum body lengths per second (Max. bl/s) were computed by dividing velocity by the body length of the models (57 cm). The time required to move one body length (tbl) and half of one body length (tbl/2) was computed for each model (1: Nautilus-like cruising thrust; 2: Nautilus-like cruising thrust scaled by the higher mantle cavity ratio of Sepia; 3: Nautilus-like peak thrust, coated and uncoated with hydrophobic silicone spray).

Model	a (V asymptote cm/s)	b	Max. bl/s	tbl (s)	tbl/2 (s)	
1	49.89 ± 0.72	0.4827 ± 0.0116	0.875 ± 0.013	2.633 ± 0.048	1.755 ± 0.032	
2	59.79 ± 0.58	0.6476 ± 0.0111	1.049 ± 0.010	2.101 ± 0.026	1.395 ± 0.018	
3 (uncoated)	162.2 ± 3.10	1.490 ± 0.045	2.846 ± 0.054	0.826 ± 0.019	0.552 ± 0.013	
3 (coated)	139.2 ± 1.50	1.970 ± 0.040	2.442 ± 0.026	0.815 ± 0.012	0.535 ± 0.008	

Modeling vertical escape maneuvers in orthocones

The time required to move one body length (tbl) or half a body length (tbl/2) under each thrust scenario was computed to determine the minimum distance (D) required to dodge a horizontally moving predator (Tables 4 and 5; Fig. 6). With higher thrust values (e.g., peak Nautilus-like thrust) these cephalopods could potentially thwart predator attacks from some relatively faster predators. A successful dodge, however, depends on predator maneuverability and burst swimming duration (Maresh et al., 2004). If D is much larger than the body length of the predator, it could easily adjust its trajectory in the vertical direction and ultimately catch the relatively slower orthocone.

Table 5 Predator evasion potential of orthocone cephalopods using mostly extant predators as analogues.

Dodges are considered successful (bold numbers) when the minimum distance required to start jetting (D) is less than the body length of a predator (Lp) moving at some incident velocity (Vp). The subscripts in D values refer to different thrust scenarios in the models (1: Nautilus-like cruising thrust; 2: Nautilus-like cruising thrust scaled by the higher mantle cavity ratio of Sepia; 3uc: Nautilus-like peak thrust with no coating; 3c: Nautilus-like peak thrust, coated in hydrophobic silicone spray). The velocity of Platecarpus (denoted by *) is only an estimate of metabolically optimal velocity (Motani, 2002), therefore critical/lunge velocity should be much higher.

					Moving one body length (57 cm)	Moving 1/2 body length (28.5 cm)	
Species	Common name	Lp (m)	Vp (m/s)	Reference	D1 (m)	D2 (m)	D3uc (m)	D3c (m)	D1 (m)	D2 (m)	D3uc (m)	D3c (m)	
Platecarpus	Mosasaur	4	0.38*	Motani, 2002	1.00	0.80	0.31	0.31	0.67	0.53	0.21	0.20	
Megaptera novaeangliae	Humpback whale	12.7	5	Segre et al., 2020	13.17	10.51	4.13	4.08	8.78	6.98	2.76	2.68	
Crocodylus porosus	Saltwater crocodile	5	8	Benga et al., 2010	21.06	16.81	6.61	6.52	14.04	11.16	4.42	4.28	
Delphinus delphis	Short-beaked common dolphin	1.8	8	Tanaka et al., 2019	21.06	16.81	6.61	6.52	14.04	11.16	4.42	4.28	
Stenella attenuata	Pantropical spotted dolphin	1.86	11	Tanaka et al., 2019	28.96	23.11	9.09	8.97	19.31	15.35	6.07	5.89	
Isurus oxyrincus	Shortfin mako shark	2.1	19	Fernandez-Waid et al., 2019	50.03	39.92	15.69	15.49	33.35	26.51	10.49	10.17	

Figure 6 Diagram of vertical predator escape and related terms.

The time required to move one body length (tbl) or half a body length (tbl/2) was computed for each velocity profile (V; Fig. 5). This time was multiplied by the velocity of various predators (Vp) to compute the minimum distance required to start jetting (D). A dodge was considered successful if D is less than the length of the predator (Lp). The image of the mosasaur (Platecarpus) was created from the outline inferred by Lindgren et al. (2010).

Discussion

The modeled ammonoid species, Baculites compressus, provides new context for the swimming capabilities, ecology, and adaptive value of the orthocone morphotype. Maintaining model hydrostatics in a chaotic, real-world setting demonstrates that movement is well constrained despite transient flow conditions surrounding the model (due to model rocking, minor ambient currents, and acceleration from a static initial condition; Figs. 3–5). Aspects of these properties likely varied between orthoconic cephalopods with increasingly different ribbing intensity, curvature, whorl section shape, degree of taper, and size. Furthermore, these cephalopods likely assumed diverse life habits during their ~420 Myr intermittent range, inferred by their disparity and occurrences in variable facies (Kennedy & Cobban, 1976; Wright, Callomon & Howarth, 1996; Kröger, Servais & Zhang, 2009). However, the present experiments reveal first-order constraints for the ubiquitous representatives of this persistent morphotype. Furthermore, a range of model buoyancies (thrusts) simulate possible modes of locomotion that are relevant to a broad range of orthocone taxa. The vertical movement potential revealed by the current experiments is applied further to model different scenarios involving predator evasion.

Vertical movement potential

The vertically-streamlined shape of orthoconic cephalopods offers several advantages for movement in the vertical direction. High hydrostatic stability would not have allowed these living cephalopods to considerably deviate from a vertical life habit (Peterman et al., 2019) unless they had sufficient counterweights to reduce stability (i.e., cameral and endosiphuncular deposits of some nautiloids; Peterman, Barton & Yacobucci, 2019). Stability acts passively and reduces energy expenditure in all ectocochleates cephalopods, regardless of conch morphology. However, the benefits of high stability in orthocones would come at the cost of maneuverability (Weihs, 2002; Fish, Hurley & Costa, 2003; Webb, 2005; Fish & Holzman, 2019). When thrust is applied in the upward direction, living orthocones would have maintained a vertical orientation with negligible deviation (<8° resulting from tilted starting positions, but generally <4° in the models; Fig. 4). The minor disturbances caused by removing the release mechanism (Fig. 3) demonstrate that slow translation in the horizontal directions can occur from weak external currents in the water column. This behavior also suggests that horizontal thrust would allow slow horizontal movement, which still could have been functional for the location of a mate or capture of slower prey items. Vertical migration, however, would have been much less expensive due to the vertical streamlining of the conch. Very low input thrust can yield high velocities relative to other ectocochleates of the same mass (Fig. 5; Tables 3 and 4). If certain orthoconic cephalopods underwent similar vertical migration patterns to extant nautilids (Ward et al., 1984; Ward, 1987; O’Dor et al., 1993; Dunstan, Ward & Marshall, 2011), they would require lower thrusts than those experimented upon to leisurely rise and fall in the water column. These diurnal vertical movements allow feeding at relatively shallower depths during the night to avoid predation (Dunstan, Ward & Marshall, 2011; Kaartvedt et al., 1996). This feeding tactic may have been more beneficial for ectocochleates in response to more abundant visual predators after the advent of the Devonian nekton revolution (Klug et al., 2010). For baculitids, there is evidence of both demersal behavior and somewhat higher occupation of the water column (Tsujita & Westermann, 1998; Landman, Cobban & Larson, 2012; Landman et al., 2018; Rowe et al., 2020). Perhaps these ammonoids were able to assume either of these lifestyles, depending on the taxon or available resources. However, it should be noted that isotope values reported for Baculites (Ferguson et al., 2019) are not comparable to vertical migration ranges recorded in Nautilus shell material (Linzmeier et al., 2016).

A slightly negatively buoyant condition would be easier for the living orthocones to manage (like extant nautilids; Ward & Martin, 1978), and represents the more conservative speculation. The downward-facing soft body may have prevented orthocones from efficiently directing a water jet upward, to counteract positive buoyancy, which may be problematic in the event of shell loss due to predation. This scenario may have been managed by ammonoid orthocones, if their complex septa improved buoyancy regulation (i.e., chamber refilling potential; Daniel et al., 1997; Peterman et al., 2021b). If complex ammonoid sutures (Peterman et al., 2021b) or larger siphuncles in some nautiloids (e.g., actinocerids, endocerids; Kröger, 2003) improved buoyancy management for these lifestyles, perhaps the magnitude of negative buoyancy could be adjusted for improved downward movement. The low metabolic cost of upward movement from reduced hydrodynamic drag would apply to downward movement as well. A very small surplus in mass (~0.5 g, ~0.26% of organismal mass; Ward & Martin, 1978) would allow orthocones to slowly drift downwards. Without jetting, living orthocones would pick up speed (as high as 15 cm/s after 20 seconds for a 57 cm individual, starting <2 m above the seafloor; Fig. 5D). However, if these cephalopods were moving too fast, velocity could be managed by small, periodic jets (i.e., braking). Furthermore, they may have been able to move in the horizontal directions through weak jetting or simply by arranging the positions of the arms. This ability is suggested by the negatively buoyant experiments, which would start tilting toward the dorsum and moving in that direction when the downward velocity reached ~8 cm/s (Figs. 3D, 4D, 5D). This behavior is likely due to the shape of the whorl section of Baculites compressus, which is wider toward the dorsum and narrower toward the venter (similar to the teardrop-shaped cross-section of an airfoil). This classic shape would have lower pressure differentials with a rounded leading edge compared to other directions (Maxim, 1896). Therefore, orthocones with whorl sections similar to Baculites compressus may have had a slight drag reduction when moving in the dorsal direction, compared to other lateral directions. Sinking may have provided a low-cost feeding strategy because incident fluid would continuously move toward the mouth (compare Peterman et al., 2021a). The reported growth direction of a cirripede attached to the baculitid, Sciponoceras (Hauschke, Schöllmann & Keupp, 2011) may have resulted from this direction of movement rather than aperture-forward, horizontal swimming. Similar epizoan growth directions on orthoconic nautiloids (Baird, Brett & Frey, 1989) may indicate that these cephalopods frequented this mode of locomotion as well. Slowly sinking toward the benthos may have even qualified as a pounce (especially for early Paleozoic orthocones), catching even slower prey items by surprise (e.g., trilobites, gastropods, other cephalopods, etc.; Alexander, 1986; Landman & Davis, 1988; Frey, 1989; Ebbestad & Peel, 1997; Westermann, 1998; Brett & Walker, 2002; Kröger, 2004, 2011).

Vertical escape tactics of orthoconic cephalopods

During the ~420 Myr range of orthoconic cephalopods, the predatory landscape changed dramatically. From the Early/Middle Ordovician to the Devonian nekton revolution, large orthocones themselves were the among the dominant predators (Brett & Walker, 2002; Walker & Brett, 2002; Kröger & Zhang, 2009; Kröger, Servais & Zhang, 2009; Klug et al., 2010). After this event, larger, faster, and speed-efficient, nektic predators would have imposed new pressure on ectocochleate cephalopods (Klug et al., 2010, 2017). These changes are reflected in the evolutionary trend of increased coiling in early ammonoids, and would have continued to influence predator-prey interactions for both persisting nautiloid groups and orthoconic ammonoids at different points in time. The high velocities (among ammonoids) in the experiments (Fig. 5C) suggest that vertical evasion tactics may have been feasible for some orthocones, provided that they have similar propulsive capabilities to modern Nautilus.

Nautiloids and ammonoids likely were eaten by other cephalopods and crustaceans throughout the Paleozoic and Mesozoic (Landman & Waage, 1986; Klug, 2007; Kröger, Servais & Zhang, 2009; Hoffmann et al., 2019). Reports of predation are scarcer for nautiloids compared to ammonoids, especially regarding orthocones (Mapes & Chaffin, 2003). However, predator-prey interactions can be inferred from specific predators through time. Other cephalopods and large arthropods would have likely assumed higher predatory roles during the early Paleozoic (Kröger, 2004, 2011). After the Devonian nekton revolution began, predation of nautiloids would have intensified from durophagous and piscivorous gnathostome fishes (e.g., placoderms, sharks, and other jawed fishes; Seuss et al., 2011) due to their superior size, maneuverability, and feeding capabilities. After the Devonian mass extinction, sharks, holocephalans, and bony fishes would have served higher-tier predatory roles. Large marine reptiles (e.g., plesiosaurs, ichthyosaurs, mosasaurs, etc.) diversified and were among the dominant components of marine ecosystems during the Mesozoic (Stubbs & Benton, 2016). Orthoconic ammonoids, primarily baculitids in the Cretaceous, have well-documented paleopathologies. Damage frequently occurs near the aperture at various growth stages (Klinger & Kennedy, 2001; Kennedy, Cobban & Klinger, 2002), likely resulting from pycnodontid fish, coleoids (Kennedy, Cobban & Klinger, 2002), and/or crustaceans (Keupp, 2012). Fatal injuries caused by mosasaurs are relatively common for Baculites as well (Kauffman, 1990; Westermann, 1996; Tsujita & Westermann, 2001). Quick upward jetting could allow easy escape from a benthic or demersal predator (e.g., crustaceans and some cephalopods). However, thwarting attacks from the relatively quicker nektic predators would require specific escape maneuvers.

The likelihood of escape would have increased if a somewhat stationary orthocone waited to jet away from a horizontally moving predator until the last possible moment (Fig. 7A). Otherwise the predator could simply adjust its trajectory and catch up to the slower, vertically moving orthocone (Fig. 7B). Within the time it takes for an orthocone to move some percentage of its body length (100% or 50%), a successful simulated dodge occurs when the distance between a predator (moving at its maximum velocity) and the orthocone is less than the body length of the predator (Fig. 6; Table 5). The condition was chosen because turning radius generally increases with velocity and body length (Maresh et al., 2004), thus reducing maneuverability and vertical correction. It should be noted that marine predators have greatly diversified in locomotor capabilities and feeding behavior through time (Brett & Walker, 2002; Stubbs & Benton, 2016; Klug et al., 2017), so this approach only represents a very general model. Furthermore, repetitive attacks from predators are not considered. Perhaps a single successful dodge would suffice if neighboring prey were captured instead, or within poorly lit waters. While this approach is simple, it suggests that orthoconic cephalopods had a fighting chance of surviving the attacks of some larger predators (using extant marine predators as analogues). Larger marine predators with similar size and speed to some modern cetaceans and saltwater crocodiles may have been outmaneuvered in some cases (Table 5). However, predators with the speed and maneuverability of modern dolphins or some sharks would have been difficult to evade (Table 5). While many aspects of predation behavior are neglected in this model, it should be noted that the vertical orientation of orthocones may have made it difficult for some vertebrate predators to attack because they would have to rotate their heads or entire bodies ~90° to bite down on the flanks of the shell. For small, quick, highly maneuverable predators (e.g., pycnodontid fish), perhaps it was more favorable for an orthoconic cephalopod to hide in its shell rather than attempting to vertically escape. Therefore, vertically escaping from larger predators that mark certain death is likely a last resort for orthoconic cephalopods, which normally assume low-energy lifestyles (Chamberlain, 1993; Mutvei, 2002; Rowe et al., 2020; Hoffmann et al., 2021).

Figure 7 Scenarios involving successful dodging (A) and unsuccessful dodging (B).

The cruising predator first notices the prey (i), then begins to accelerate (ii). After closing in (iii), the predator makes its final lunge for the prey (iv). Cones surrounding the predator indicate hypothetical turning radiuses. For a successful dodge, the orthocone cephalopod must wait until the last possible moment or else the incoming predator could adjust its vertical trajectory. The image of the mosasaur (Platecarpus) was created from the outline inferred by Lindgren et al. (2010).

Paleoecological interpretations

The low-cost vertical movement and momentarily rapid escape inferred for orthocones by the current experiments better constrain the ecology of these cephalopods. Existing morphological and ecological information can be used along with these inferred locomotive capabilities to better elucidate the life habits of these ubiquitous animals.

The largely unknown soft body characteristics of orthoconic cephalopods could contribute to differences in available thrust and velocities. If the propulsion of ammonoid orthocones is more similar to coleoids (Jacobs & Landman, 1993), closer relatives than nautiloids (Kröger, Vinther & Fuchs, 2011), they may have been able to produce larger thrusts than the highest thrust value used in the current experiments (peak Nautilus-like thrust). However, it is unknown how confinement of the soft body in a rigid shell would reduce propulsive power compared to coleoids which can hyperinflate their mantle cavities (Anderson & Demont, 2000). Furthermore, the lack of well-preserved soft body material (Klug, Riegraf & Lehmann, 2012; Klug & Lehmann, 2015; but see Klug et al., 2021) complicates homologization of muscles inferred by scars on the shell (Kennedy, Cobban & Klinger, 2002; Doguzhaeva & Mapes, 2015) to the musculature of extant coleoids or nautiloids. Therefore, the thrust values in the current study pose a wide range of somewhat conservative thrust estimates for ammonoids. For orthoconic nautiloids (orthocerids, endocerids, and actincocerids, among others), muscle scars are different from those of extant Nautilus and tarphycerids which suggests that certain orthocone clades may have been weaker swimmers compared to coiled nautiloid clades (Mutvei, 2002; King & Evans, 2019). In this case, the Nautilus-like peak thrust of the current experiments may be too liberal. However, the magnitude of propulsive differences due to differences in muscle scar size and extent are unclear in the absence of soft part preservation or modern orthocone analogues.

Though far from orthocone analogues, extant shrimpfish (a.k.a. razorfish) assume downward-facing vertical orientations. However, in contrast to orthoconic cephalopods, they can pitch their bodies into horizontal orientations during swimming due to their comparatively lower hydrostatic stability (Fish & Holzman, 2019). Furthermore, these fish can turn about their longitudinal axis with ease due to their low moments of inertia (Atz, 1962; Fish & Holzman, 2019). While this behavior was not investigated in the current study, this property seems likely for orthocones as well, based on their mass distribution and general shape. However, rather than using various fins, orthocones would be limited to swimming with their arms and hyponome near the aperture (if the soft body morphology permitted), which is likely not as effective. The transverse cross-section of shrimpfish (Fish & Holzman, 2019) is also similar to many baculitid ammonoids (see Larson et al., 1997), which infers drag would be reduced in the dorsal direction. Although this horizontal mode of locomotion in orthocones would not be very efficient due to the higher hydrodynamic drag relative to vertical movement, and their low source of jet thrust (i.e., the thrust angle; Okamoto, 1996; Peterman et al., 2019; Peterman, Mikami & Inoue, 2020). The vertical orientation of shrimpfish is thought to be associated with camouflage (Fish & Holzman, 2019); a function unlikely for comparatively larger orthocones, and those in habitats lacking structure in which to hide.

Exceptional preservation has been reported more commonly for baculitid ammonoids than any other orthocone (Klug, Riegraf & Lehmann, 2012; Klug & Lehmann, 2015), allowing more specific interpretations of their life habit. The preservation of large putative eye capsules and the presence of lateral sinuses at their apertures suggest enhanced predator detection capabilities (Nilsson et al., 2012). The large ventral rostrum on many baculitid shells may have restricted thrust in the ventral direction, suggesting limited horizontal mobility (unless the hyponome could bend around the rostrum using the lateral sinuses). These features, along with the hydrodynamic properties inferred by the current experiments support a life habit of slowly searching for planktic prey, while maintaining the option for somewhat rapid vertical escape from large predators. For baculitids, this life habit is consistent with well-preserved mouthparts (aptychi and radulae) that support planktivory (Landman, Larson & Cobban, 2007; Kruta et al., 2009; Kruta et al., 2011). A low energy lifestyle of searching for small, planktic prey can be facilitated by low-cost vertical movement or from relatively sedentary, demersal behavior (e.g., around methane seeps teeming with life; Landman et al., 2018; Rowe et al., 2020).

The vertical motility of orthoconic cephalopods is higher relative to planispiral ammonoids, based on their differences in hydrostatic centers (thrust angle; Okamoto, 1996; Peterman et al., 2019; Peterman, Mikami & Inoue, 2020) and vertical streamlining (Westermann, 1996). This advantage suggests they could have occupied a distinct niche among ectocochleates. The availability of this niche may reflect suitable ecological opportunities arising from dynamic ecological conditions (e.g., predatory pressure, food resources; Cecca, 1997; Reboulet, Giraud & Proux, 2005). Similarly, dynamic environmental conditions (e.g., sea level rise, microhabitat availability; Yacobucci, 2015) also drive selection and speciation in ectocochleates, which may have increased the availability of this niche in newly-formed epeiric seas.

Our results offer interpretations for the adaptive value of orthoconic cephalopods and potential evolutionary drivers behind their iterative recurrence in the fossil record. Due to the many uncertainties regarding predator evasion and the putative low-energy lifestyle of orthocones (Chamberlain, 1993; Mutvei, 2002; Rowe et al., 2020; Hoffmann et al., 2021), we suggest their low-cost vertical motility and unique hydrostatic properties to be a primary driver of their evolution and success. While orthocones may have maintained the ability to dodge benthic or nektic predators, these capabilities are likely restricted to taxa without substantial conch curvature. Instead, transitional forms (e.g., cyrtocones; Hoffmann et al., 2021) show gradual changes in hydrostatic properties; static orientations with progressively downward-facing apertures and thrust angles improved for upward locomotion. These trends throughout orthocone ammonoid evolution are generally the inverse of those observed during ammonoid origination (Klug & Korn, 2004; Kröger & Mapes, 2007; Monnet, De Baets & Klug, 2011; Monnet, Klug & De Baets, 2015). Furthermore, these transitional forms would not likely facilitate rapid movement.

Orthocones with a higher degree of morphological disparity from our target species, Baculites compressus, could have progressively differing hydrostatic and hydrodynamic properties that constrained their life habits and swimming capabilities. Conch size, curvature, whorl section shape, ornamentation, body chamber proportion, and internal characteristics (e.g., septal/siphuncular morphology and counterweights) vary within this morphotype. The fluted flanks of some orthocones and variable whorl section shapes (e.g., B. grandis, B. obtrusus, etc.) could have influenced hydrodynamics in some way. Furthermore, some orthocones are curved (grading into the cyrtocone morphotype), which could pose difficulties in steering rapid, upward maneuvers. In order to understand the roles and consequences of these characteristics, their full spectra would need to be investigated with future experiments. Furthermore, stability-reducing counterweights (Peterman, Barton & Yacobucci, 2019) in the phragmocones of certain taxa must be investigated further in a dynamic setting to determine their potential influences on posture while swimming.

Conclusions

The high hydrostatic stability of orthoconic cephalopods without cameral deposits (Peterman et al., 2019; Peterman, Barton & Yacobucci, 2019) would have strictly constrained the life habits of these animals. They would have been confined to vertical orientations without the capacity to substantially modify them. These properties raise questions about the modes of life, functional morphology, and adaptive value of the orthocone morphotype. The hydrodynamics inferred by 3D-printed models of the baculitid, Baculites compressus, suggest that vertical movement was well constrained for orthocones (Figs. 3 and 5). High hydrostatic stability prevents rocking during movement, even with somewhat variable starting angles (Fig. 4). It seems orthocones are adapted to both upward-vertical movement through active locomotion and passive downward-vertical movement at very little cost. High velocities relative to other ectocochleates of similar size suggest that low hydrodynamic drag is incurred by movement in the upwards direction. Therefore, these living cephalopods required very low energy to vertically migrate in the water column (probably at velocities lower than those in the current study; Fig. 5). Slight negative buoyancy (like extant nautilids, Ward & Martin, 1978) would have allowed these cephalopods do slowly sink after jet thrust is suspended. This condition would have allowed low-energy movement and feeding for vertical migrants while also providing suitable speeds to pounce on benthic prey from above. While these cephalopods likely assumed low energy lifestyles, Nautilus-like peak thrust would have given orthocones a fighting chance at vertically escaping attacks by larger predators. Although coiled ectocochleate cephalopods flourished from the Devonian to end of the Cretaceous, orthoconic cephalopods retained adaptive value during this time as low-energy vertical migrants, with the potential to serve as periodic escape artists.

Supplemental Information

Supplemental Information 1 Raw data from 3D motion tracking.

Velocity, position, and angle data for each trial of motion tracking.

Click here for additional data file.

We appreciate the help of Emma Janusz and Mark Weiss for accommodating our experiments at the University of Utah pool (the Crimson Lagoon). We would also like to thank K. De Baets, B. Kröger, A. Pohle, and R. Lemanis for their thoughtful comments and constructive reviews of the manuscript.

Additional Information and Declarations

Competing Interests

Author Contributions

Data Availability

The authors declare that they have no competing interests.

David J. Peterman conceived and designed the experiments, performed the experiments, analyzed the data, prepared figures and/or tables, authored or reviewed drafts of the paper, and approved the final draft.

Kathleen A. Ritterbush conceived and designed the experiments, authored or reviewed drafts of the paper, and approved the final draft.

The following information was supplied regarding data availability:

The raw motion tracking data is available in the Supplemental File.

The external 3D model of Baculites compressus is available at MorphoSource: DOI 10.17602/M2/M359359.

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
