# Peer review of "Vertical escape tactics and movement potential of orthoconic cephalopods"

_PeerJ, doi:10.7717/peerj.11797_

## Round 0.1 · original submission · Minor Revisions

This is a very nice contribution on the possible adaptive value of a straight shell in externally cephalopods which is also underlined by the minor revisions proposed by the reviewers. I would love to see this published, but there are still some crucial points which need to be addressed before publication:

1) Scope: the general title suggests quite broad-sweeping conclusions although you only focus on a single taxon with a compressed whorl section (see also comment by reviewer 2). This is no critique on your study which is nicely designed and executed, but rather about the way it is presented/promoted. Also, the taxon is well chosen as it had a coiled ancestor. There would be two options to resolve this issue – a) make clearer from the start (in introduction) that you focus particularly on baculitids and that additional experiments are necessary to test the full bandwidth of the capabilities of orthoconic cephalopods (in discussion); b) add additional orthoconic taxa with different morphologies (which might be outside the scope of your current study). As far as I know – some baculitids have also been interpreted to be slightly curved or would this rather be a taphonomic phenomenon? Either way – it might still be interesting to test or at least discuss how slight curvature would impact your experiments as well as how gradual re-evolution of an orthoconic shape might impact your interpretations.

2) Paleozoic orthoconic cephalopods: I feel adding additional references on Paleozoic orthoconic nautiloids and their diversity would be crucial (both in case of scenario 1a or 1b as described under point 1; see particularly reviewer 2). I have the impression that orthoconic forms often co-occur with coiled forms, but that at least orthoconic morphologies were comparatively more diverse and abundant in the Paleozoic and re-appeared in the Mesozoic but never in the diversity and disparity available in the Paleozoic. Both aspects should likely be discussed. I cannot shake the feeling that having a low-energetic lifestyle became possible again during certain parts of the Mesozoic related with productivity and widespread habitats. I also do not see such a vertical escape strategy being very effective on the long run (not without efficient forward movement or “sinking” or in the presence of smart predators which could back around; compare Reviewers 2 and 3).

3) Escape tactics and Evolutionary pressure: I am not entirely convinced that predation pressure were major evolutionary factors driving the evolution of orthoconic morphologies on their own, but I do agree that vertical escape tactics might have been a major side effect of having an orthoconic morphology. I feel that the development of a low-energetic lifestyle rather than predatory pressure might also have been a key driving force here or at least some kind of ecological release (compare Reviewer 3) – at least during the re-appearance of this morphologies in the Mesozoic. The situation might have been more complex and diverse during the Paleozoic (see Reviewer 2). Also, not necessarily all orthoconic forms had a vertical orientation, particularly in the Paleozoic. I am also wondering how the (in)efficient vertical movement of coiled forms would have been in comparison. It would be worth to discuss these hypotheses more broadly. Encrustation patterns might also help to support some of your hypotheses (see Baird et al. 1989 as mentioned by Reviewer 1)

4) Ecological saturation: you mention ecological saturation a couple of times, but it is not clear how that would work exactly as orthoconic and coiled forms often seem to co-occur in these habitats (compare Reviewer 3). Maybe it is rather the abundance in resources and prey items which makes this strategy successful in the Mesozoic despite predation.

5) Supplementary material: It would be worth considering uploading (some of) the video material of your experiments at least as supplementary material.

Please address all other points raised by me and the reviewers including those listed in the annotated pdfs in addition to these points.

I look forward to seeing your revised manuscript.

·

Basic reporting

This manuscript is part of a series of groundbreaking papers recently published by the first author and cooperators about mode of life and the functionality of the shell architecture of ectocochleate cephalopods. The research question, followed in the manuscript, is what, if at all, is the adaptive value of the orthocone cephalopod shell. Repeated, independent re-evolution of this shell form suggests an adaptive value. The method, how this hypothesis is answered combines traditional ideas, namely the reconstruction of a physical shell model, with up-to date computational methods and statistical tests. In combination these methods give surprising results and new insights.
The original idea of an advantage of orthoconic cephalopods by rapid vertical escape from pelagic predators, which in contrast often have a mode of life optimized for rapid horizontal hunting is compelling and the authors support their hypothesis with a robust set of data. Also, the results clearly support the hypothesis that orthococonic shells give an advantage for low-cost vertical movement.
All underlying data supporting their hypothesis and have been provided in detail (including raw data) by and statistical error margins are well documented.
The conclusions are well stated and the manuscript, generally is written in a clear language without redundancy and unnecessary detail. The figures are clear, relevant and of high quality. The figure captures are easily readable.
I have only one very minor suggestion:
Baird et al (1989) described syn-vivo epizoans on Palaeozoic orthocones. The growth directions of the bryozoans, described therein could be used to support the arguments of the authors (Baird, G. C., Brett, C. E., & Frey, R. C. (1989). Hitchhiking epizoans on orthoconic cephalopods: preliminary review of the evidence and its implications. Senckenbergiana lethaea, 69(5-6), 439-465.

Experimental design

no comment

Validity of the findings

no comment

Additional comments

At line 324 "However, this is .." it is not clear what the "this" refers to.

·

Basic reporting

First, I want to congratulate the authors for this very interesting article. The manuscript is well written and presents an innovative approach. I am looking forward to it being published. During the revision, I came across some issues which need to be addressed.

Some of the references could be expanded. In particular, some additional references for nautiloid cephalopods should be cited apart from the Treatise and Kröger, Servais & Zhang 2009. In addition, some important literature on hydrostatics and locomotion of orthoconic cephalopods is missing and should at least be cited somewhere or discussed. Some suggestions:

Schmidt 1930, Über die Bewegungsweise der Schalencephalopoden. Paläontologische Zeitschrift 12, 194–208 [in German].
Teichert 1933, Der Bau der actinoceroiden Cephalopoden. Palaeontographica, Abt. A 78, 111–230 [in German].
Flower 1955, Cameral deposits in orthoconic nautiloids. Geological Magazine 92, 89–103.
Crick 1988, Buoyancy regulation and macroevolution in nautiloid cephalopods. Senckenbergiana lethaea 69, 13–42.
Chamberlain 1993, Locomotion in ancient seas: constraint and opportunity in cephalopod adaptive design. Geobios 15, 49–61.

Further literature suggestions in the relevant comments below.

Experimental design

The manuscript presents an innovative approach for testing hypotheses on the mode of life of orthoconic cephalopods. Orthocones are highly abundant in some localities during the Palaeozoic and the same morphotype reappeared during the Mesozoic in heteromorph ammonites. Thus, identifying the locomotory constraints of orthocones is important for our understanding of these past ecosystems. However, this is complicated by a lack of modern analogues, and hence, their swimming capabilities and ecological roles are poorly understood. This manuscript therefore offers a refreshing new perspective on this topic, including new tools to test hypotheses.

I have nothing to add to the experiments themselves, except that you may need to explain better the rationale behind testing a sinking model, since this would not be relevant for the vertical escape hypothesis. I understand why you did it, but I still cannot quite see the connection of this part of the results to the conclusions. What alternative results could have been expected other than slowly accelerating sinking in a stable position? Would have different results changed your conclusions in any way? If not, why do it in the first place?

Perhaps it would be great to upload part of the video footage somewhere and link it in the article?

I also want to add some thoughts, which you may incorporate into the manuscript, use them as inspiration for future studies, or simply discard them. I am not sure if further experiments are feasible, they may be out of scope for the current study. Right now, you tested three models with considerable positive buoyancy and one model with only very slight negative buoyancy. From these experiments, I find it difficult to interpret whether orthocones were better adapted to upwards or downwards movement. If vertical escape tactics were one of the primary drivers for orthoconic conch shapes, then I would expect the shell to be optimized for rapid upward movement. However, if downward movement is more stable, then I would expect that escape tactics are only a secondary advantage. You mention that downward movement would be difficult because of the position of the hyponome. However, you already suggested active buoyancy regulation in ammonites (Peterman et al. 2021), and it has also been proposed in various nautiloid groups with orthoconic conchs and very large siphuncles (e.g., actinocerids and endocerids, see Kröger 2003, The size of the siphuncle in cephalopod evolution, Senckenbergiana lethaea 83, 39–52). Thus, they were maybe able to achieve higher degrees of negative buoyancy and consequently, higher accelerations in downward direction? Obviously, this depends on the rates of chamber filling, which is another issue. One could test whether orthocones are better adapted to downward or upward movement, perhaps at different speeds. Intuitively, I would say downward is more optimal, though it may depend on the shape of the soft body and would thus be difficult to test. What is also not tested (for obvious reasons I suppose) is sinking or rising at slow but constant speed, as the animals may have used small bursts of the hyponome as a brake in either direction. This may be out of scope for the current article, but perhaps provides some ideas for future studies!

Validity of the findings

I am not sure if vertical escape movements were the main drivers for the evolution of orthoconic conch shapes, though you make a compelling case that this was at least possible. It appears to me that vertical escape movements would only work in a completely straight conch, but how would evolution go from a tightly coiled to a straight conch then? Either there would be a direct “switch” between coiled and orthoconic (which seems unlikely to me), or successive evolutionary stages of uncoiling, but those cyrtoconic intermediate stages probably could not employ vertical escape tactics, so there must be another evolutionary advantage. In addition, many orthocones are not completely straight, but at least slightly curved, though often this is only visible in very long fragments. Would these curved conchs not impede vertical escape tactics? Perhaps feeding strategies were more important, e.g., filter feeders moving up and down or pouncing on benthic prey as you mention already. In any case, orthocones show such a diversity in external and internal structures during the Palaeozoic, that they must have played different ecological roles (also indicated by their facies distributions). Thus, my recommendation is that you discuss in more detail other advantages of an energy efficient vertical movement capability in orthocones. Perhaps make it clear that vertical escape tactics were likely not the only evolutionary reason for orthoconic conchs, but you show that despite the seemingly clumsy and awkward vertical orientation (this seems to be one of the main reasons why older publications categorically excluded this possibility!), they were not as slow or unmaneuverable as one would think. Furthermore, I recommend that you point out more clearly at the beginning that you are only testing a very specific conch shape, namely a slender smooth orthocone with a compressed cross section and without counterweight. Especially the part with the compressed cross section should be mentioned somewhere, as at least to me – being relatively unfamiliar with Baculites – it was not clear at the beginning and I had to look it up. There are so many other shell characteristics that may have an influence, such as cross section shape, expansion rate, annulations, ribs, longitudinal ornament, apertural modifications, conch curvature, mass distribution (i.e., deposits), etc. Perhaps you could try to assess the potential impact of these characters on your conclusions, thus allowing for a more differentiated picture, which of your findings can likely be generalized for most orthocones and which cannot. Obviously, testing all this with additional experiments would be out of scope of your study. Nevertheless, this is a great first step, opening up new opportunities to identify the influence of these traits on hydrodynamics of orthocones and our understanding of the palaeoecological role of this morphotype!

Additional comments

Multiple times throughout the manuscript, you state that the orientation of orthocones was vertical or near vertical, citing Peterman et al. (2019). While I liked the latter article a lot, I do not think that it can be concluded from this alone that all orthocones had vertical orientations. For example, you assume a very long body chamber of at least 33% in your other paper, but from my experience, many (at least Palaeozoic) orthocones have shorter body chambers (e.g., the endocerid Anthoceras vaginatum has an estimated body chamber length between 16% to 20% of the total conch length, see Kröger 2012, The “Vaginaten”: the dominant cephalopods of the Baltoscandian Mid Ordovician endocerid limestone, GFF 134, 115–132). Shorter body chambers would allow more apical chambers to be filled with cameral or endosiphuncular deposits, potentially adding enough ballast for a horizontal orientation. A hypothetical vertical orientation would also make it impossible for endocerids to empty their chambers, because they have extremely long septal necks (thus, they could not accumulate enough gas in the phragmocone to become buoyant, unless they had a different mode of chamber formation than Nautilus). In most endocerid species, the siphuncle touches the ventral shell wall, and thus, a horizontal orientation would conveniently allow for complete chamber emptying. Anyway, I just wanted to highlight that we cannot rule out horizontal modes of life for at least some orthocones. You mention already colour markings and cameral/endosiphuncular deposits, but you need to be careful that you do not contradict yourselves: Sometimes you mention possible horizontal alternative orientations, but on other occasions you appear to suggest that significant deviations from a vertical position were not possible at all. Therefore, I propose that you try to be more consistent on this issue. For example, you could mention potential horizontal orientations already in the introduction when talking about orientation first. Then, you could make it clear that you refer to vertical (unballasted?) orthocones during the remainder of the text.

Further minor comments:

Title and elsewhere: The term “orthocone” is a noun, while “orthoconic” is the corresponding adjective (see Flower 1964 or Treatise). Hence, I suggest using either “orthoconic cephalopods” or “orthocones”, depending on context. For the title, the former would be appropriate. Please check throughout the manuscript.

Line 50: To a reader unfamiliar with cephalopods, it may appear as if a tightly coiled conch evolved only during the Devonian – however, this happened multiple times as well: Tarphycerida in the Early Ordovician (e.g., Teichert et al. 1964), independently the Uranoceratidae in the late Ordovician (Kröger 2013: The cephalopods of the Boda Limestone, Late Ordovician, of Dalarna, Sweden, European Journal of Taxonomy 41, 1–110), the Lituitida in the Middle Ordovician (particularly Cyclolituites, Sweet 1958, The Middle Ordovician of the Oslo region, Norway, Norsk Geologisk Tidsskrift 38, 1–178; note that the group is now thought to be derived from the Orthocerida rather than the Tarphycerida), Hercoceratidae in the Early Devonian (Turek 2007, Systematic position and variability of the Devonian nautiloids Hercoceras and Ptenoceras from the Prague Basin (Czech Republic), Bulletin of Geosciences 82, 1–10) and of course the Nautilida in the Devonian (or Silurian?), which may originated from one of the above groups or independently from somewhere within the Orthocerida (Kröger, Vinther & Fuchs 2011). Thus, orthoconic and coiled forms coexisted essentially continuously starting from the Early Ordovician, but probably occupied different habitats and ecological roles.

Line 97: This is not so clear - earlier publications have also seen them as bottom dwellers (e.g., Teichert 1967, Major features of cephalopod evolution. University of Kansas Department of Geology, Special Publication 2, 162–210) or active swimmers (e.g., Holland 1987, The nautiloid cephalopods: a strange success, Journal of the Geological Society, London 144, 1–15). One recent publication interpreted endocerids as pelagic suspension feeders (Mironenko 2020, Endocerids: suspension feeding nautiloids?, Historical Biology 32, 281–289), although the latter results are somewhat speculative. In general, the mode of life of orthoconic cephalopods is poorly understood and palaeobiological studies are lacking. It seems likely that different lifestyles existed - this is also indicated by the diverse siphuncular structures found in Palaeozoic orthocones.

Line 246: I suggest you use meters instead of foot to describe the depth of the pool.

Line 376: Conch curvature could also be added to this list, for very slightly cyrtoconic forms.

Line 429: Just a thought: wouldn’t the airfoil-like cross section help the animal to move in horizontal (i.e. forward/dorsal) direction?

Line 460: There are also well documented cases of sublethal injuries in Palaeozoic endocerids (Kröger 2011, Size matters – Analysis of shell repair scars in endocerid cephalopods, Fossil Record 14, 109–118) and orthocerids (Kröger 2004, Large shell injuries in Middle Ordovician Orthocerida (Nautiloidea, Cephalopoda), GFF 126, 311–316), as well as Devonian orthocerids, bactritids and ammonoids (Klug 2007, Sublethal injuries in Early Devonian cephalopod shells from Morocco, Acta Palaeontologica Polonica 52, 749–759). Note that in the latter study, there was no difference found between the frequencies of injuries in straight and coiled conchs.

Line 471: Why would the predator not just turn around to follow the orthocone? At some point, it will hit the surface without further room to escape. Perhaps these evasion tactics were more useful against benthic or demersal predators?

Line 505: You may also cite Klug et al. (2021, Failed prey or peculiar necrolysis? Isolated ammonite soft body from the Late Jurassic of Eichstätt (Germany) with complete digestive tract and male reproductive organs, Swiss Journal of Palaeontology 140, 3).

Line 506: See Doguzhaeva & Mapes (2015, The Body Chamber Length Variations and Muscle and Mantle Attachments in Ammonoids, in: Klug et al. (eds.), Ammonoid Paleobiology: From Anatomy to Ecology, Topics in Geobiology 43, 545–584) for details on ammonoid muscle attachments.

Line 509: Just as a clarification, there are also orthocones within other nautiloid groups, e.g., Ellesmerocerida (even in the Cambrian, such as Ectenolites), Cyrtocerinida (e.g., Bathmoceras), Oncocerida (Jovellaniidae) and Bisonocerida (e.g., Allotrioceras). The Riocerida and Dissidocerida are also predominantly orthoconic. Thus, there is a variety of with diverse siphuncular morphologies (though you are correct that where known, orthocones seem to have relatively small retractor muscles, see also King & Evans 2019, High-level classification of the nautiloid cephalopods: a proposal
for the revision of the Treatise Part K, Swiss Journal of Palaeontology, 138, 65–85).

Fig. 4-5: Please adjust the position of the subfigure labels (A-D) to be consistent across figures. Additionally, please use a sans-serif font in the axis labels of Fig. 5. Lastly, to avoid confusion, it may be a good idea to place the legend in Fig. 5C at the bottom end of the figures, since it applies to all subfigures (even if two colors only appear in 5C).

·

Basic reporting

The manuscript is well written and structured. The figures are high-quality and easy to read.

Experimental design

The design of the experiments seems well suited to addressing the major goal of establishing limitations on the vertical movement of orthoconic shells.

Validity of the findings

The conclusions regarding vertical acceleration and passive downward movement relative to the potential ecology seem justified although the conclusions about the ability to dodge predators seems highly speculative.

Additional comments

The authors have conducted an interesting study into the hydrodynamics of accelerating orthocones. By utilizing a clever system of weights and cavities within a 3D print of a Baculites, they were able to compare vertical movement speed between several different acceleration types. The MS is well written with well-made figures and is easy to understand. My major criticism of this MS is that all of this good work is seemingly done in service of a story that his heavy on speculation and not very convincing. This focus on predation and the evolution of orthocones as a function of vertical escape is never really explained. Overall, I would suggest minor revisions since the science presented here seems sound but the discussion is a bit weak.

In the discussion, the authors mention past research that relates the evolution of increased coiling in early ammonoids to the evolution of quick, nektonic predators. But the authors suggest that the opposite trend, from spiral to straight shells is also, to some degree, a function of nektonic predators. Maybe this innate high stability of these forms permitting a low-energy lifestyle is the driving force here and perhaps the lack of maneuverability seen in orthocones relative to spiral shells is an indication that predation wasn’t a significant evolutionary pressure? In this case the speed in upward movement is just due to the conical shape that is not necessarily an adaptation to movement as it is a necessary consequence of the accretionary growth of the animal.

All of this speculation about dodging predators seems to assume that a predator could not reorient itself in the water and simply chase the orthocone whose only escape vector is up, which would bring it closer to the surface-water interface where it would likely be fairly easy prey. Then factoring into this, the fact that the actual distance traveled from the beginning of acceleration in real life would not be constant, as in the model since the buoyant force is constant, but would “pulsed” due to the jet. How far could it travel on a single pulse? Is it possible to very roughly estimate this?

A few other minor things:
Is there a kind of missed opportunity here in not simulating the downward movement of the animal as you did the upward movement? Of course this would have been an overestimate of downward movement since hooking the hyponome around the aperture would displace the thrust vector from the center of rotation but even as a “best-case” scenario it might have been interesting in constraining possible speeds.

I believe the stability index you use (Eq. 3) is the one Okamoto suggested in his chapter: Theoretical modeling of ammonoid morphology in Ammonoid Paleobiology. Maybe cite this.

On line 229 you mention the 3D prints have watertight voids. Are these not printed with a support material? If so how did you remove it?

You mention that only a relatively small amount of mass, ~0.26% of the organisms mass (does this just mean soft-body mass?) is required for a negatively buoyant condition. Do you know how large the crop of Baculites was? If it was filled do you think it would have been enough mass to cause the animal to sink?

Line 429, you say that Baculites is similar in shape to an airfoil. I don’t really follow what you mean here. Are you talking about the dorsal/ventral wall thickness?

Line 487, you say the animal would have to rotate it’s head 90 degrees but since they are in water couldn’t it also just rotate it’s entire body 90 degrees and attack with that orientation?

Lines 530-540 A couple of times in the MS you mention ecological saturation. I think this needs to be explained a bit more. Planktonic and demersal habitats are fairly commonly suggested for planispiral forms as well. What exactly is the niche the orthocones are filling that other forms could not?

Line 533 What is the vertical motility of planispiral ammonoids? You didn’t test this and I don’t see a real comparison from the literature. Is there another paper you’re referring to here?

---

## Round 0.2 · Minor Revisions

Thank for addressing all of our suggestions. Thank you for also providing a link to one of the videos. I consider the manuscript as good as accepted. However, I found some additional very minor aspects I would like to address before publication. As they might involve citing some additional references - this would be consistent with a minor revisions decision.

---

## Round 0.3 · accepted · Accept

Thank you for addressing these final suggestions. Looking forward to seeing this published.